**Large drainages from short-lived glacial lakes in the Teskey Range,**
**Tien Shan Mountains, Central Asia**
Chiyuki Narama[1], Mirlan Daiyrov[1,2], Murataly Duishonakunov[3], Takeo Tadono[4],
Hayato Satoh[1,5], Andreas Kääb[6], Jinro Ukita[1], Kanatbek Abdrakhmatov[7]
1) Department of Environmental Science, Niigata University, Niigata, Japan
2) Central-Asian Institute for Applied Geosciences (CAIAG), Bishkek, Kyrgyzstan
3) Department of Physical Geography, Kyrgyz National University
4) Japan Aerospace Exploration Agency (JAXA), Tsukuba, Japan
5) Kokusai Kogyo Co., Ltd, Tokyo.
6) Department of Geoscience, University of Oslo, Norway
7) Institute of Seismology, Kyrgyz Academy of Science, Kyrgyzstan
Keywords: short-lived glacial lake, depressions, debris flow, Tien Shan
**Abstract**
Four large drainages from glacial lakes occurred during 2006–2014 in the western
Teskey Range, Kyrgyzstan. These floods caused extensive damages, killing people and
livestock as well as destroying property and crops. Using satellite data analysis and field
surveys of this area, we find that the water volume that drained at Kashkasuu glacial
lake in 2006 was 198,000 $m^3$, that at western Zyndan lake in 2008 was 437,000 $m^3$, that
at Jeruy lake in 2013 was 163,000 $m^3$, and that at Karateke lake in 2014 was 169,000 $m^3$.
Due to their subsurface outlet, we refer to these short-lived glacial lakes as the
"tunnel-type", a type that drastically grows and drains over a few months. From spring
to early summer, such a lake either appears, or in some cases, significantly expands
from an existing lake (but non-stationary), and then drains during summer. Our field
surveys show that the short-lived lakes form when an ice tunnel through a
debris-landform gets blocked. The blocking is caused either by the freezing of stored
water inside the tunnel during winter or by the collapse of ice and debris around the ice
tunnel. The draining occurs then through an opened-up ice tunnel during summer. The
growth–drain cycle can repeat when the ice-tunnel closure behaves like that of typical
supraglacial lakes on debris-covered glaciers. We argue here that the geomorphological
characteristics under which such short-lived glacial lakes appear are (i) a
debris-landform containing ice (ice-cored moraine complex), (ii) a depression with
water supply on a debris-landform as a potential lake-basin, and (iii) no visible surface
outflow channel from the depression, indicating the existence of an ice tunnel. Applying
these characteristics, we examine 60 depressions ($> 0.01$ km$^2$) in the study region and
identify here 53 of them that may become short-lived glacial lakes, with 33 of these
having a potential drainage exceeding 10 m$^3$/s in peak discharge.
**1. Introduction**
The northern Tien Shan in Kyrgyzstan, Central Asia, contains many small
glacial lakes at glacier fronts. Compared to many large proglacial lakes in the eastern
Himalayas that exceed 0.1 km$^2$ (Komori et al., 2004; Nagai et al., 2017), 74% of the
lakes in the study region had area extents of less than 0.01 km$^2$ in 2014 (Narama et al.,
2015). Nevertheless, in recent decades, rapid drainage from such lakes in the Central
Asian Mountains has caused severe damage for residents in nearby mountain villages
(Kubrushko and Staviskiy, 1978; Kubrushko and Shatravin, 1982; Narama et al., 2009;
Mergili and Schneider, 2013). More recently, catastrophic damage occurred in 1998
from an outburst of the Archa–Bashy glacial lake in the Alay Range of the Gissar–Alay
region. This small lake, which had formed on a debris landform at the glacier front,
suddenly released over 50,000 m$^3$ of water. Although the volume of released water was
relatively small, the flood killed more than 100 residents along the river in Shahimardan
village in Uzbekistan (UNEP, 2007), demonstrating that flood volume alone is an
insufficient indicator of damage potential.
In a similar event on 7 August 2002 in the Shahdara Valley, Pamir, Tajikistan, a
320,000 m$^3$ drainage from a small lake caused a mud-flow that buried the Dasht village
on the alluvial fan and killed 25 people (Mergili et al., 2012). In the northern Tien Shan,
a drainage occurred from the western Zyndan glacial lake in the Teskey Range on 24
July 2008 (Narama et al., 2010a). The latter event discharged 437,000 m$^3$ of water,
causing extensive damage, killing three people and many livestock as well as destroying
a bridge, a road, two houses, crops and an important fish-hatchery. These lakes are a
type of short-lived glacier lake (non-stationary, but existing only over a short period)
that appear and discharge within a few months or one year (Narama et al., 2010a;
Mergili et al., 2013).
Unlike the supraglacial cases (e.g., in the Italian Alps; Haeberli, et al., 2002;
Harrsion et al. 2015), the Central-Asian cases are a different type of short-lived glacial
lake that appear at the glacier front on debris-landforms with buried ice. Monitoring of
such lakes is complicated due to their sudden and short appearance (Narama et al.,
2010a). Also, their drainage through ice tunnels differs from that from many other
glacial-lake-outburst floods (GLOFs) in the eastern Himalayas (Bhutan and eastern
Nepal), which discharge through the failure of a moraine dam (Yamada and Sharma,
1993; Watanabe and Rothacher, 1996; Komori et al., 2012). In addition, the growth
period of a short-lived lake also differs from the large proglacial lakes that have
continued to expand since the 1950s–1960s in the eastern Himalayas (Ageta et al.,
2000).

In the western Teskey Range of the Tien Shan mountains, several large floods
occurred from a glacial lake at the Angisay Glacier in 1974, 1975, and 1980 (Kubrushko
and Staviskiy, 1978; Kubrushko and Shatravin, 1982). Since then, large drainages have
occurred at Kashkasuu glacial lake in 2006, at western Zyndan lake in 2008, at Jeruy
lake in 2013, and at Karateke lake in 2014. Among them, the drainage from the western
Zyndan lake was examined in Narama et al. (2010a). To understand the
geomorphological characteristics of this type of short-lived glacial lake, we examine all
four of these above recent floods as well as depressions with short-lived lakes using
field-surveys and satellite data. To help decrease the damage from such glacier-related
disasters, we assess the locations and volumes of these depressions and current glacial
lakes. In addition, we discuss the geomorphological characteristics that lead to
short-lived glacial lakes.

**2. Study area**
Along the southern shoreline of Lake Issyk-Kul in Kyrgyzstan, Central Asia,
lie the Teskey Range, Tien Shan mountains (Fig. 1). The Tien Shan is a reactivated area
of Paleozoic deformation. Although this region had a low-relief surface following the
Paleozoic orogenies, Late Cenozoic deformation has resulted in this surface being
warped across a series of mountain ranges cored by crystalline basement and previously
deformed Paleozoic sedimentary and metamorphic rocks (Abdrakhmatov et al., 2001;
Burgette et al., 2017). Here, surrounding Lake Issyk-Kul, lies the Issyk-Kul basin,
bound to the south by the actively growing Teskey Range. The Teskey Range has a
ridgeline at 4800–3700 m asl above small alpine glaciers.
Most precipitation here occurs in May–July, when the weakened Siberian High
allows moisture to arrive from the west (Aizen et al., 1995). In general, the northern
Tien Shan (outer ranges of the Tien Shan) blocks moisture carried by the Westerlies,
causing larger annual precipitation amounts in the Pskem, Talas, Kyrgyz, Ili, and
Kungöy Ranges, compared to the Teskey Range (Narama et al., 2010b) . In this range,
the average annual precipitation ranges from 363 mm (1981-1999) in the western part
(Karakujur station; 3000 m asl), to 247 mm (1981-1999) in the central part (Tien Shan
station; 3600 m asl), to 597 mm (1981-1987) in the eastern part (Chong-Kyzylsuu; 2550
m asl).

Glacier shrinkage in the outer and inner ranges also varies significantly
throughout the Tien Shan (Narama et al., 2010b). The glacier area has decreased less in
the west than in the east (Narama et al., 2006; Katuzov and Shahgedanova, 2009).

The population and villages are distributed over the northern part of the Teskey
Range. There, villagers use the large alluvial fans at the mountain piedmont as pasturage
or agricultural fields.

**3. Methods**
**3.1 Field surveys**
In the study area of the western Teskey Range, we investigated glacial lakes
and three recent (2006–2014) large drainages based on field surveys (2007–2016) and
satellite data analysis. The drainages include those from the Kashkasuu, Jeruy, and
Karateke glacial lakes shown in Fig. 1. We visited 25 glacial lakes including lakes that
caused a large drainage, and investigated landforms (distance and location of ice tunnel,
depression of lake basin) and lake levels after drainage using a Trimble GeoExplorer
6000 (cm-precision edition) and a Leica GPS 900. We obtained a positional accuracy of
within 20 cm through differential post-processing of Trimble GPS data using the
Kyrgyz GPS reference station. To estimate the water volume of current lakes, we
measured water depths in 10 current lakes in the area using an inflatable boat
(PVL-260) and a fish finder with GPS(LOWRANCE HDS-5; Fig. 1). In the
downstream zones of Jeruy and Karateke lakes, we investigated clast diameter,
sedimentary facies, extent of flood deposit, and their eroded channels. In addition, we
interviewed residents of Jeruy village about local floods.

**3.2 Satellite data analysis**
We investigated the evolution of the Kashkasuu, Jeruy, and Karateke lakes
using the Advanced Land Observing Satellite (ALOS) with the Panchromatic
Remote-sensing Instrument for Stereo Mapping (PRISM; 2.5-m resolution), as well as
the ALOS AVNIR-2 (10-m resolution), Landsat7 ETM+, and Landsat8 OLI data. The
ALOS and Landsat images were fused, and pan-sharpened images using the PCI
Geomatica software were used to estimate glacial lake areas by manual mapping of the
glacial lake boundaries. We also estimated the water volumes after drainage at
Kashkasuu, Jeruy and Karateke lakes using ALOS PRISM digital surface models
(DSMs) taken on 19 Nov. 2007 and 10 Aug. 2010. The PRISM DSMs were processed
by JAXA EORC as a high-level product. The standard deviations of the PRISM DSM
height errors (PRISM DSM without Ground Control Points, GCPs, minus reference
DSM) are between 4.9 and 8.7 m (Takaku and Tadono, 2009; Tadono et al., 2012).
In general, a short-lived glacial lake appears at a depression (shallow hollow)
on the debris landforms at glacier fronts. To estimate the location and maximum volume
of a potential lake-basin, we used a water-filling model to extract depressions on the
debris-landforms at glacier fronts. The model used PRISM DSM data taken on 17 Sep.
2007, 19 Nov. 2007, 28 Apr. 2010, 10 Aug. 2010, 10 Nov. 2010, and 27 Nov. 2010. We
set 0.01 km$^2$ as the minimum depression size, because recent drainages with damages
are caused from short-lived lakes exceeding 0.01 km$^2$. To test the accuracy of the
water-filling model, we compare its result to an estimate based on GPS data from the
western Zyndan lake before its drainage (24 July 2008). As shown in Fig. 2, the GPS
data along the shoreline (ice line) of the western Zyndan lake before drainage coincides
well with the extracted outline of the depression from the model. Existing lakes are also
extracted using ALOS Advanced Visible and Near Infrared Radiometer-2 (AVNIR-2)
images taken on 17 Sep. 2007, 3, 8 Sep. 2008, 10 Aug. 2010.
For assessment of floods, it is of significant advantage to anticipate the flow
type and map the landform of a valley mouth reached by the flood because the latter
determines the range and form of the debris spread. To distinguish debris-flow types
(i.e., debris flow vs. water flood), we estimated the erodible channel distance as the
distance over which the channel has an angle exceeding 10° using NASA's shuttle radar
topography mission digital elevation model (SRTM) DEM (30-m resolution). (This data
covers regions outside the ALOS DSM region that we plan to use for further study.)    In
addition, we used satellite data to classify the lowland landforms in 23 valleys in the
northern part of the western Teskey Range into valley landform and alluvial fan.

**4. Results**
**4.1. Evolution of three short-lived glacial lakes**
In the following, we consider the area changes of the Kashkasuu lake in 2006,
Jeruy lake in 2013, and Karateke lake in 2014. (The western Zyndan lake is described in
detail in Narama et al., 2010a). Figure 3 shows their changes observed from satellite
images, Fig. 1 shows their locations. Kashkasuu lake in the southern part of the Teskey
Range directly contacts a glacier, so we call it a "glacier-contact" type. In Fig. 3A, we
show its area increases on 21 June 2005 (compared to 6 Sept. 2004, not shown), and
remains nearly the same on 23 May 2006. It grows until 26 July 2006, expanding to
0.025 km$^2$, but the lake shrinks again to 0.004 km$^2$ on 11 August 2006. Thus, lake water
has discharged between 26 July and 11 August 2006. Based on the lake area on 26 July
2006 and the PRISM DSM, we estimate that 198,000 m$^3$ of water volume discharged.
To the northwest of Kashkasuu lies Jeruy glacial lake. Images in Fig. 3B show
this lake to be not recognizable on 18 May 2013, but clearly visible by 19 June 2013.
By 6 August 2013, it has grown to 0.033 km$^2$ with an estimated volume of 163,000 m$^3$.
The lake, which has glacier contact, drains on 15 August 2013, but some water remains
on 23 September.
Nearby and to the east lies Karateke lake. This lake is without glacier contact
and located on a debris-landform at the glacier front. Figure 3C shows the lake area to
be only 0.001 km$^2$ on 5 May 2014, but expanding to 0.02 km$^2$ on 30 June 2014, and
then decreasing to 0.015 km$^2$ on 16 July immediately before drainage on 17 July 2014.
After drainage, its area becomes 0.0016 km$^2$. During this drainage, 169,000 m$^3$ of water
was discharged.
These three lakes, as well as the western Zyndan lake that discharged 437,000
m$^3$ on 24 July 2008 causing a large flood (Narama et al., 2010a), all appear in May,
grow rapidly in June and July (Fig. 4), then discharge between mid-July and
mid-August. Thus, all four lakes are examples of a "short-lived glacial lake" that
suddenly appears and grows during two or three months, with drainage occurring in the
summer. Considering the growth of the lakes, the Kashkasuu glacial lake appeared and
remained from the previous year, but then grew suddenly. But the Karateke, Jeruy, and
western Zyndan lakes evolved and grew from an initially empty basin during the same
year.

**4.2 Geomorphological evidence of drainage from the short-lived lakes**
To better understand the behavior of the lakes, particularly their drainage, we
investigated adjacent landforms in a field survey. At the Kashkasuu lake, in 2007, a
debris landform containing ice was found at the glacier front. The ice-rich debris
landforms are also called moraine complex (Shatravin, 2007; Janský et al., 2010; Bolch
et al., 2014). The debris-landform, composed of debris and ice, remained from glacier
shrinkage. The Kashukasuu lake expanded on the large depression (hollow) with glacier
contact. No surface channels were visible, but we observed a subsurface channel that
developed inside of the debris landform. It was a 300-m-long ice tunnel with a
water-stream from the lake to the tunnel outlet. The lake water discharged through this
ice tunnel between 26 July and 11 August 2006 (Fig. 3A). From August 2006 (before
drainage) to September 2007, GPS data shows the lake level dropping by 10 m. This
large drainage damaged the mountain road and a bridge along the Uchemchek River.
We observed exposed ice and ice tunnels on similar debris-landforms in front
of the Jeruy and Karateke Glaciers (Fig. 5A, B). Both debris-landforms contain buried
ice. Jeruy lake appeared on the depression of a basin with glacier contact. Karateke lake
also formed at an empty depression, but without glacier contact. For the Karateke lake,
meltwater from the glacier terminus flows into the depression. But for the outlets of
both lakes, we observed no visible surface outflow channel from either depression.
However, we found the Jeruy depression to have a 250-m-long ice tunnel and the
Karateke to have a 500-m-long one. For the Karateke lake, the ice tunnel is 4-m wide at
the entry point of the debris landform (Fig. 5C). The middle point of the ice tunnel is
5-m deep (Fig. 5D).

Our field survey thus indicates that lake water from the Kashkasuu, Jeruy, and

Karateke lakes discharged through ice tunnels inside of debris-landforms, as was found
previously also for the western Zyndan lake (Narama et al., 2010a). In these
debris-landforms, there are no visible surface outflow channels, and most meltwater
from the glacier flows through an ice tunnel. Hence, we consider these short-lived
glacial lakes as "tunnel-type" to distinguish them from those that discharge through
different mechanisms (e.g., dam failure, surface channel blockage).

**4.3 Flood deposits and landforms**

Regarding the flood deposits from the lakes studied, the Jeruy and Karateke

Valleys are located side by side (Fig. 1), but they produce different flood types and
damage. The flood deposits from the Jeruy drainage consist of matrix-support deposits
of clasts of mostly 0.20–0.30 m diameter but also including boulders of 1–3 m diameter
(Fig. 6A). From an interview of a local resident of Jeruy village we confirmed that the
flood velocity of the Jeruy drainage was slow on the alluvial fan. The flood stream from
the Jeruy glacial lake separated into two routes on the large alluvial fan and did not flow
along the present water stream. On the alluvial fan, the flood caused a bridge collapse
and damaged an irrigation channel, a road, many tombs, an agriculture field, and the
fence of a house. On the other hand, flood deposits from Karateke lake consist of
clast-supported deposits with large boulders of 1–2 m diameter (Fig. 6B). Flood
deposits are limited to the riverbed. Damages from the Karateke flood were limited to
two bridges along the river. The flood deposits of the western Zyndan were similar to
the Karateke deposits.

For the Jeruy Valley (Fig. 6C), the uneroded flat riverbed section in the upper

part is short and the erosion section is long. In contrast, for the Karateke Valley the
upper part consists of a flat valley with only a short highly eroded section (Fig. 6D). The
different erosion distances are related to the valley landforms in the upper valley part

from past glaciation. When a steep slope starts at the end of a flat valley, the flood-wave is able to gain debris, transforming to a debris-flow. As an indication of a flood wave , grass flattened by water in the riverbed after drainage from the western Zyndan lake (Narama et al., 2010a).

The degree of entrainment and the resulting changes in water and debris content influence the velocity of the flow (Breien et al., 2008). In the flat valley section below the Karateke lake, we also observed flattened grass along the river. Although the Jeruy drainage has a long eroded section which can indicate a fast flow, the flood here did not include many large boulders. The flow had relatively slow velocity on the alluvial fan. In contrast, dense debris flows could be quite fast. During experiments in the Ili Range of the northern Tien Shan, Kazakhstan, the mean density of highly mobile debris flow reached 2200 kg/m$^3$ with water content below 10% (Baimoldaev and Vinohodov, 2007; Evans and Delaney, 2015). When the eroded material is dry, entrainment produces a high concentration of solids in the slurry with associated increases in the viscosity, cohesion, and friction, all of which could reduce the mobility (Breien et al., 2008).

## 4.4. Volume size of existing lakes and depressions

To estimate the water volume and basin form of current glacial lakes, we measured the depths and geolocations of 10 lakes using an inflatable boat and fish finder with GPS. All 10 lakes were less than 30-m deep. Profiles of three of them are shown in Fig. 7. Lakes in the study area are small. Of the 160 glacial lakes over 0.001 km$^2$ in the Teskey Range, 68% of them are less than 0.01 km$^2$ (Narama et al., 2015). The resulting profiles of the lake-basins at glacier fronts are asymmetric as shown in Fig. 7B and C, with greater depth and steeper slope at the glacier terminus side. We found a submerged moraine at the lake bottom of the eastern Zyndan lake. Such a moraine prevents a complete discharge of all lake water, but most observed lakes had no such internal barriers.

The short-lived glacial lake type studied here appears at a depression on a debris landform containing buried ice. To find the locations of such depressions in the northern part of the western Teskey Range, we used the PRISM DSMs (2.5-m resolution) to measure the distribution and volume of depressions of potential lake-basins. In total, we found 60 depressions exceeding 0.01 km$^2$. A short-lived glacial lake can appear at such a depression only if it receives sufficient meltwater. Thus, we distinguished the depressions as those with glacier contact and those without glacier contact (Fig. 8). Of the 60 depressions, 38 (i.e., 63%) of the depressions had glacier

contact in which meltwater can inflow from glacier termini.
The depressions without glacier contact are of the "water accumulation" and
"non-accumulation" types. The depressions of water accumulation type can get
meltwater from the glacier because the depression is connected to it via one or more
subsurface channels. In contrast, the non-accumulation type is not connected to a water
channel and cannot get substantial amounts of water within short time. We found 22
depressions of the water accumulation type, and each may become a short-lived lake
such as the Karateke lake (Fig. 3C). In addition, we determined whether or not the
depression had a surface outflow channel. Among the 60 depressions, 7 depressions had
a surface outflow channel and thus cannot hold a short-lived lake of the tunnel-type
studied here.
The relationship between area and volume of the 10 measured lakes agrees
with those found previously. In the plot of Fig. 9, we also show the four large drainages
of Kashkasuu, w-Zyndan, Jeruy, and Karateke glacial lakes, as well as depressions in
this study area, and six lakes from previous studies in the Kyrgyz and Ili Ranges
(personal communication of I. Severskiy; Janský et al., 2010). The regression line
formula between area and volume of existing lakes was calculated using only
measurement data (present and previous studies), and was then used to estimate the
water volume of current lakes. The larger-area lakes have volumes above the
regression-line fit because those depressions tend to have steeper sides. Although we
could use a second-order fit to better fit these points, our subsequent analyses below all
use lakes with areas below 0.05 km$^2$. Thus, we use the simpler first-order relation.

**5. Discussion**
**5.1 Geomorphological characteristics of tunnel-type, short-lived glacial lakes**
The field surveys of the four short-lived lakes revealed that, in each case, water
discharged through an ice tunnel inside an ice-rich debris landform. Our satellite
observations of these short-lived glacial lakes show them to appear as a small pond in
May and to expand suddenly in June–July. The field surveys showed this behavior to be
due to i) the blockage and closure of ice tunnels, and ii) rapid melting of snow and ice in
the upstream area. The ice tunnels become blocked due to freezing of stored water
during winter or deposition of ice and debris by tunnel collapse. Later, their drainage
between the end of July and mid-August occurred when their ice tunnel opened, due to
subsurface ice melting or evacuation of debris at the closure point.
This drainage process seems similar to the closure and opening (connection) of
an englacial conduit for a supraglacial lake on a debris-covered glacier (Benn et al.,
2001; Gulley and Benn, 2007; Gulley et al., 2009). Supraglacial lakes have a seasonal
variability and can be transient or recurring, depending on the connectivity to englacial
network (Miles et al., 2016, 2017; Benn et al., 2017; Narama et al., 2017). The
formation and sudden drainage of supraglacial ponds also have occurred in the
Cordillera Blanca, Peru (Emmer et al. 2015). Several large drainages from supraglacial
lakes through englacial conduits have occurred on debris-covered glaciers without a
large proglacial lake in front of them (Richardson et al., 2009; Komori et al., 2012;
Rounce et al., 2017). In north-western Nepal, a supraglacial lake that developed
temporally at a depression on a small alpine glacier caused a large drainage through an
englacial conduit (Kropáček et al., 2015).
This drainage process differs from that typical for glacial lake outburst floods
(GLOFs) in the eastern Himalayas. The Himalayan GLOF type occurs from a large
proglacial lake that has expanded for several decades but then has its moraine dam fail
(Yamada and Sharma, 1993; Ageta et al., 2000). Such a lake typically does not refill to
the same level it had before failure. However, the lake area might expand again
backwards due to glacier recession, or the water level might increase due to blockage of
an outlet channel when a large-scale failure occurs at the moraine's inner slope (Ageta et
al., 2000) or the dam opening is blocked by snow and ice (Huggel et al. 2003). After
drainage, the failure is visible as a V-shaped channel excavated across the moraine dam
(Breien et al., 2008; Komori et al., 2012).
In contrast, the short-lived glacial lake type studied here appears and expands
for a few months, and then discharges through ice tunnels. The lake appears in a
depression that develops after recent glacier recession (Narama et al., 2010a). After
drainage, vertical subsidence occurs along the subsurface channel in the debris landform.
Such a short-lived glacial lake type recurs when its ice tunnel closes, similar to that on a
supraglacial lake on a glacier or on a debris-covered glacier (Kropáček, et al., 2015;
Benn et al., 2017; Narama et al., 2017). For example, at Angisay Glacier in the Teskey
Range (Fig. 1), several floods occurred from the same glacial lake in 1974, 1975, and
1980 (Kubrushko and Staviskiy, 1978; Kubrushko and Shatravin, 1982). The repeated
floods indicate that the lake water refills at the same lake basin due to a repeated closure
of the ice-tunnel. Although existence of the lake could not be confirmed, the Ak-Say
Glacier of the Kyrgyz Range also had repeated drainages in the 1980s (Janský et al.,
2010; Zaginaev et al., 2016).

**5.2 Identifying potential lake-basins of short-lived glacial lakes**
In addition to monitoring existing lakes, our findings suggest that one should
also monitor empty depressions in which a short-lived glacial lake may form. But which
depressions should be monitored? We can rule out some depressions by examining
several features of the depression and its environment as follows.
One characteristics to rule out some cases is not having a clear source of
meltwater. That is, a short-lived glacial lake cannot appear at a depression in which
meltwater cannot inflow at substantial amounts. Among the 60 depressions ($> 0.01$ km$^2$)
we examined, 38 of them had glacier contact and thus can get meltwater directly from
glacier termini. The remaining 22 depressions had no glacier contact, but could also
accumulate water type such as the Karateke lake (Fig. 3C). As another
geomorphological feature to rule out potential hazardous cases, in the case of the basin
has outflow channels, a short-lived lake cannot storage much water for a short term. We
exclude seven basins with surface outflow channels from the depressions. As a result of
these two restrictions, 53 depressions among 60 depressions are found to be potential
basins for a tunnel-type, short-lived glacial lake. Considering now the factors that
influence drainage volume from a short-lived lake, one factor is the volume of the
depression. For example, a supraglacial lake formed in a large depression caused a large
drainage from the Tshojo Glacier in the Lunana region, Bhutan (Yamanokuchi et al.,
2009). The Karateke lake drainage had a volume of 169,000 m$^3$, the smallest of the four
large drainages we studied, and the lake had an area of 0.015 km$^2$. Thus, we recommend
to monitor depressions with area exceeding 0.01 km$^2$, also taking into account moderate
future expansion from recent glacier recession.
Other factors are the timing of the ice-tunnel opening and the melting rate of
ice and snow, both of which affect the water filling of the depressions. In the study area,
the lake water of the western Zyndan glacial lake overflowed before a large drainage on
24 July 2008. The overflow was due to a high snow/ice melting rate and a late timing of
ice-tunnel opening (Narama et al., 2010a). In contrast, an early timing of ice-tunnel
opening or a small upstream melt rate might cause only a partial discharge.
In addition, the width of the ice tunnels and distance to the closure point
determine the total stored water volume (lake plus conduits) because the closure point
may be far downstream from the lake in the ice tunnel. Thus, the drainage volume
depends on (i) volume of the depression, (ii) timing of the ice-tunnel opening, (iii) melt
rate of ice and snow, (iv) size of the ice tunnel, and (v) closure point of the ice tunnel.

**5.3 Transition to debris flow**
For lakes of tunnel type, the flood wave without moraine deposits can
transform into a debris flow where the channel gets steeper and the wall-material
erodible. The change occurs because banks of the channels in the study area are often
composed of loose material (Haeberli, 1983; Clague and Evans, 1994, Breien et al.,
2008; Evans and Delaney, 2015). The mobility of the debris flow also depends on the
type of loose erodible material. In the Ili Range of the northern Tien Shan, some cases
of drainage started with a small initial failure volume that then increased by entrainment
of material from the path, acquiring much debris from the middle of a steep mountain
slope, resulting in very large deposits that exceeded $10^6$ m$^3$ (Baimoldaev and Vinohodov,
2007; Evans and Delaney, 2015).

Two main types of debris flows occur, viscous and stony (Takahashi, 2004;
2009). For example, the Jeruy and Karateke lakes have about the same elevation (3815
and 3757 m asl), and similar maximum discharge (Qmax) values of 13.9 and 14.2 m$^3$/s,
but Jeruy's debris-flow type was a viscous flow with matrix-supported deposits (Fig.
6A), whereas Karateke's was a stony debris-flow with clast-supported deposits (Fig. 6B).
To help us understand these differences, we also investigated the debris flows that
occurred on 3 June 2009 from the Takyltor Glacier in the Kyrgyz Range.

Where does the channel erosion occur? We assume erosion occurs where net
deposition does not occur. According to Hungr et al. (1984), net deposition in a channel
starts where the channel angle is about 10° or less. Thus, we use the total distance over
which the channel exceeds 10 degrees as defining the 'erodible channel distance'. This
erodible channel distance agrees well with the actually eroded distance in the western
Zyndan lake case (Narama et al., 2010a). Although the lake elevation, maximum
discharge, and slope angles of the erosion section are about the same at the Karateke
and Jeruy Valleys, the erodible channel distances vary significantly by valley (Fig. 6C,
D).

Observations show that entrainment make debris flows more and more erosive,
resulting in a feedback effect (Breien et al., 2008). This effect partly explains the high
rate of volume increase observed in many debris flows and is probably often necessary
to achieve long runouts in subaerial flows.

We characterize potential flows by the erodible channel distance and the
estimated maximum discharge. We estimate here the maximum discharge of 60
depressions and existing lakes using the duration of discharge and $Q_{max} = 46(V/10^6)^{0.66}$
(tunnel event; Walder and Costa, 1996), with a water volume V. This formula neglects
the possible role of tunnel size in total drainage volume. For an existing lake, V is
estimated using the regression formula in Fig. 9. In the study area, the erodible channel
distances range between 166 and 6016 m, and the maximum slope gradients of the mean
erodible channel distance are 11.5–20.9°. We also characterize actual drainage events
(the above four recent floods) using these two parameters.
The results, plotted in Fig. 10, suggest a classification in which each drainage
is either a debris flow or water flood. Many short-lived lakes change from water flood
to debris flows, involving debris entrainment (stony flow or viscous flow), due to the
channel wall having looser material (including composition of material; fragmented
rock or surficial materials; Evans and Delaney, 2015). With a transition boundary at an
erodible channel distance of about 1500 m, the western Zyndan, Karateke, Jeruy,
Kashkasuu, and Takyltor floods are classified as debris flows in Fig. 10, and they have
Qmax values of 14–27 $m^3$/s. Among the 53 depressions, 33 depressions exceed 10 $m^3$/s
in possible maximum discharge (Fig. 10). But for the existing lakes, many lakes have a
Qmax below 10 $m^3$/s.
Thus, the drainage of short-lived lakes in the study region shows a transition
between water flood and debris flow. The debris entrainment along the erosion distance
can add a considerable amount of debris deposits. Depending on the situation, the debris
flow may be stony or viscous flow.   Although the influence of most debris flows from
the short-lived lakes is limited to the valley mouth or river sides on the mountain
piedmont, the deposition region may in some cases have a long runout if the fluidity
increases with distance along the flow.

**5.4 Differences in flood damage**

On the alluvial fan downstream of the Jeruy Valley, two debris-flow streams
separated from the present river channel and caused large damages to agriculture fields,
irrigation infrastructure, roads, and many tombs. In comparison, in the Karateke Valley
only two bridges were broken because the outburst stream was limited along the river.
In Shahimardan village, where a flood killed more than 100 residents (UNEP, 2007),
many residents live along the river. In the Dasht village, where a flood killed 25 people
(Mergili et al., 2012), the debris-flow covered the village on the alluvial fan.
The degree of flood damage is related to the local land-use and the landform
type (e.g., alluvial fan) at the valley mouth. During the western Zyndan lake drainage in
2008, the flood damaged a kashaal (animal cottage) on the alluvial fan (Narama et al.,
2010a). Among 23 valleys in the study area in the northern-western part of the Teskey
Range (Fig. 1), 14 valleys are a valley landform (e.g., Karateke case),   the other 9
being an alluvial fan type (e.g., Jeruy case). The drainages from the four short-lived
lakes studied here are less than 500,000 $m^3$ and their flood damages are limited along
the river or alluvial fan. As most depressions are up to 500,000 $m^3$ in this region, most
flood damages are considered to be likely limited along the river or alluvial fan at the

valley mouth. Although some large depressions have existed here, we know of no case in which a large lake had a large drainage. However, for risk mitigation, drainages from short-lived lakes should become an integral part of river basin management in the region, considering in particular depression volume, flood type, land-use, and landforms potentially affected.

**6. Conclusions**

In the western Teskey Range, recent large lake drainages have come from the tunnel-type of short-lived glacial lakes, which appear and then drain within a few months. These lakes were found to typically appear as small ponds in May, then expand suddenly in June–July due to rapid melting of ice and snow in the upstream area. The lake damming appears due to blockage and closure of ice tunnels, as a result of winter freezing of stored water, or deposition of ice and debris by tunnel collapses. The drainage then occurs between the end of July and mid-August when the ice tunnel re-opens, due to ice melting or the blocked section flushed away. Using the estimated drainage volumes from the current lakes or depressions, we argue that their flood damages will occur only at their alluvial fans or along the river at their mountain piedmont. Most drainage events led to debris-flows.

The geomorphological characteristics under which these lakes appear were found to be (i) a debris-landform including dead ice (ice-cored moraines) with the potential to form an ice tunnel, (ii) a depression ($> 0.01$ km$^2$) on the debris-landform with sufficient water supply, and (iii) no visible outflow channel from the depression, thus requiring the water to exit through an ice tunnel.

The comparably short period of a few months between appearance and drainage of the short-lived lake type studied here poses a special challenge to the application of satellite remote sensing for monitoring them. However, new satellite constellations such as Sentinel-2 (5 days repeat; Kääb et al., 2016) or the Planet cubesat constellation (daily repeat, Kääb et al., 2017) will help to detect even short-term changes. The applicability of Sentinel-1 radar data (6 days repeat) for monitoring these lakes remains to be tested (Strozzi et al., 2012). For such systematic surveillance, the type of prioritization of potentially dangerous sites as proposed here is essential. We propose an early information network based on monitoring with satellite data that informs the responsible authorities and possibly local people when a lake appears. As glacier-lake workshops in the Ladakh region of India (Ikeda et al., 2016) and in Jeruy village (study area) showed, improvement of knowledge and land-use can help reduce the impacts of large drainage floods form glacial lakes.

In the Tien Shan, depressions ($> 0.01$ km$^2$) in which water can inflow should be
monitored, just as we now monitor glacial lakes, and their potential associated hazards
considered. Lake monitoring using satellite data should proceed based on the criteria of
potential dangerous lakes outlined here such as the location and volume of the lakes and
depressions, the flood type, and landform on the mountain pediment.

**Acknowledgement**
Special thanks are due to O. Moldobekov, S. Usupbaev, C. Ormukov of Central-Asian
Institute for Applied Geosciences (CAIAG), of CAIAG, A. Aitaliev of the Ministry of
Emergency Situations of the Kyrgyz Republic, I. Severskiy of Institute of Geography,
Kazakhstan, S. Erokhin of Geological Institute of the Kyrgyz Republic, and local people.
We also thank an editor F. Catani, A. Emmer and two anonymous reviewers for valuable
comments. This study was supported by the Mitsui & Co. Environmental Fund in
2013-2015 and Grant-in-Aid for Scientific Research (C) 25350422 and (B) 16H05642
of the Ministry of Education, Culture, Sports, Science and Technology (MEXT), and
Heiwa Nakajima Research Foundation. This study used ALOS satellite image data from
ALOS Research Announcement (RA) in the framework of JAXA EORC. A. Kääb
acknowledges support by the European Research Council under the European Union's
Seventh Framework Programme (FP/2007–2013)/ERC grant agreement no.320816, and
the European Space Agency (ESA) within the Glacier_CCI (code 400010177810IAM)
and DUE GlobPermafrost projects (4000116196/15/IN-B). This work is also a
contribution to the SIU CryoJaNo project (HNP-2015/10010).

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

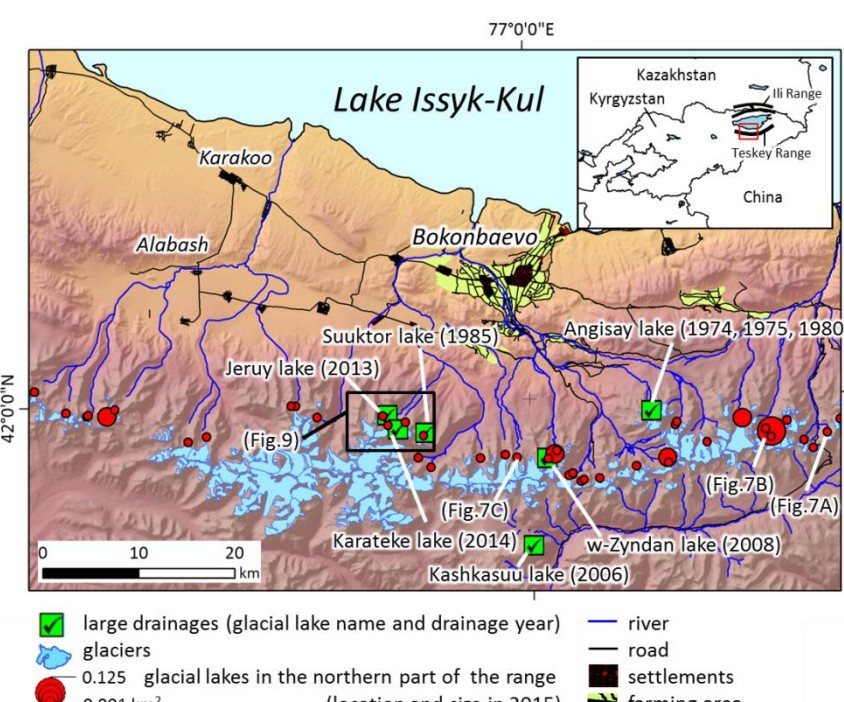


Fig. 1. The study area in the western part of the Teskey Range, Kyrgyzstan. Green boxes
show the location of large drainage events with the name and year labeled. Location and
size of red circles show locations and size of lakes in 2015.

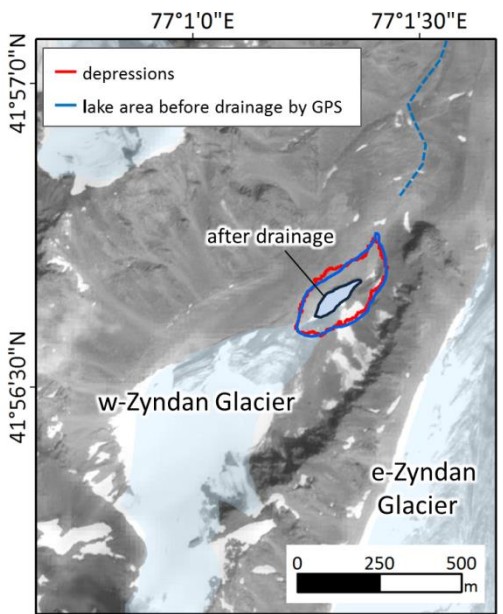

Fig. 2. Western Zyndan glacial lake at which a large drainage occurred in 2008 (location
in Fig. 1). The blue line shows the lake perimeter before drainage according to GPS
measurements in 2008. The red line shows the depression according to ALOS PRISM
DSM data.

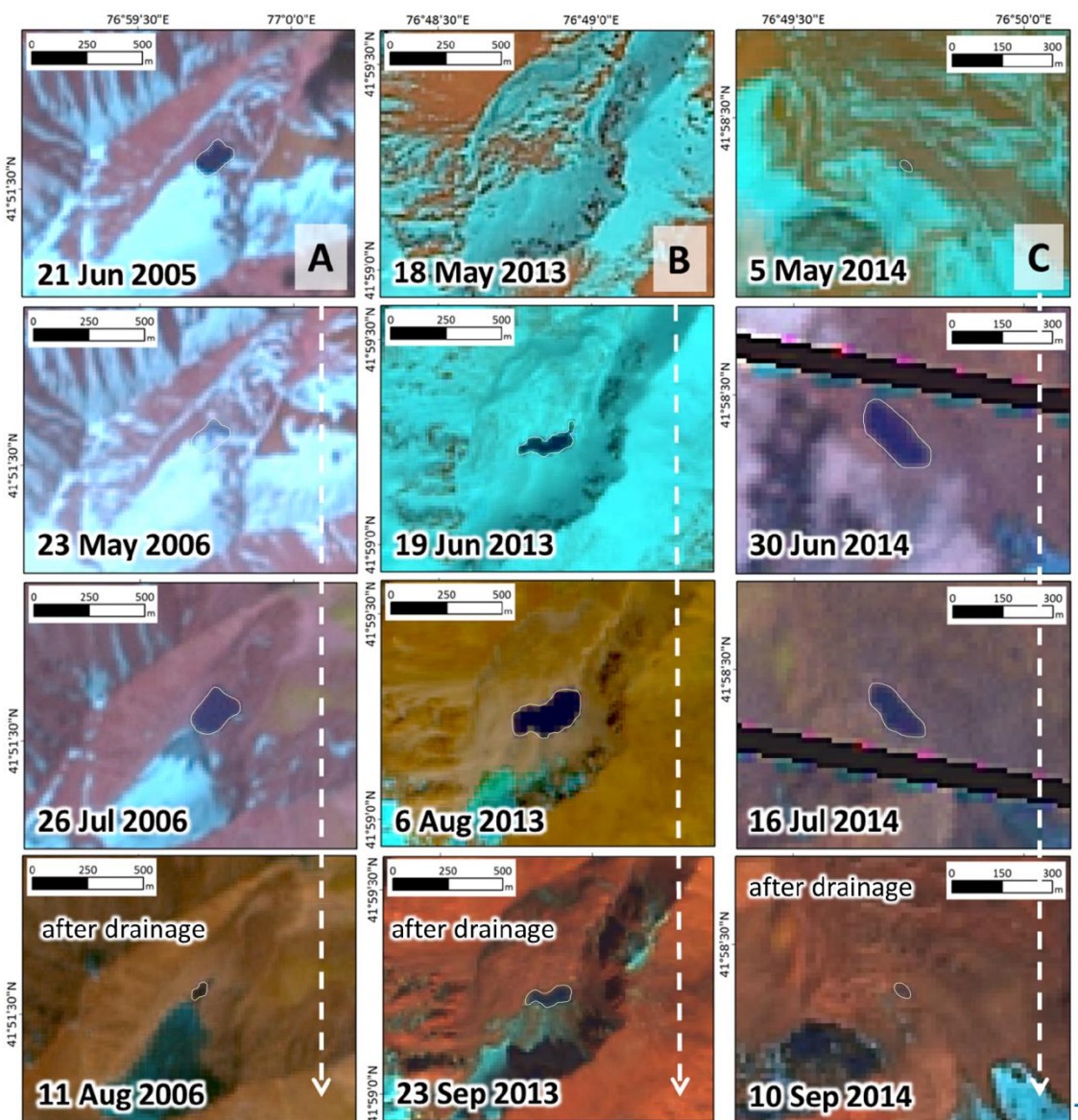


Fig. 3. Changes of three lakes. Left column (A) is Kashkasuu, middle (B) is Jeruy, and
right (C) is Karateke lake. Images are from Landsat7 ETM+ and ALOS AVNIR-2 and
PRISM data. The locations are in Fig. 1.

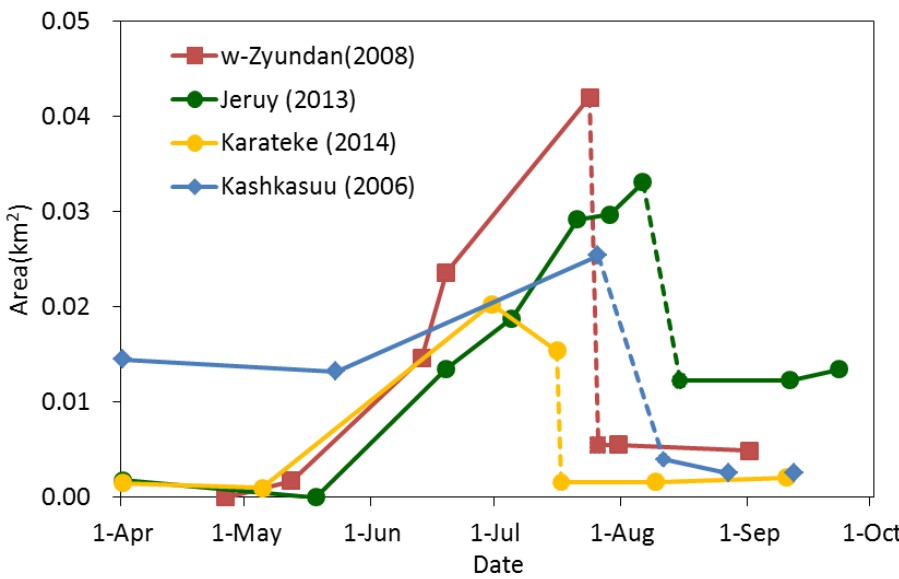


Fig. 4. Seasonal area changes of four short-lived glacier lakes. Dashed line segments
indicate drainages.

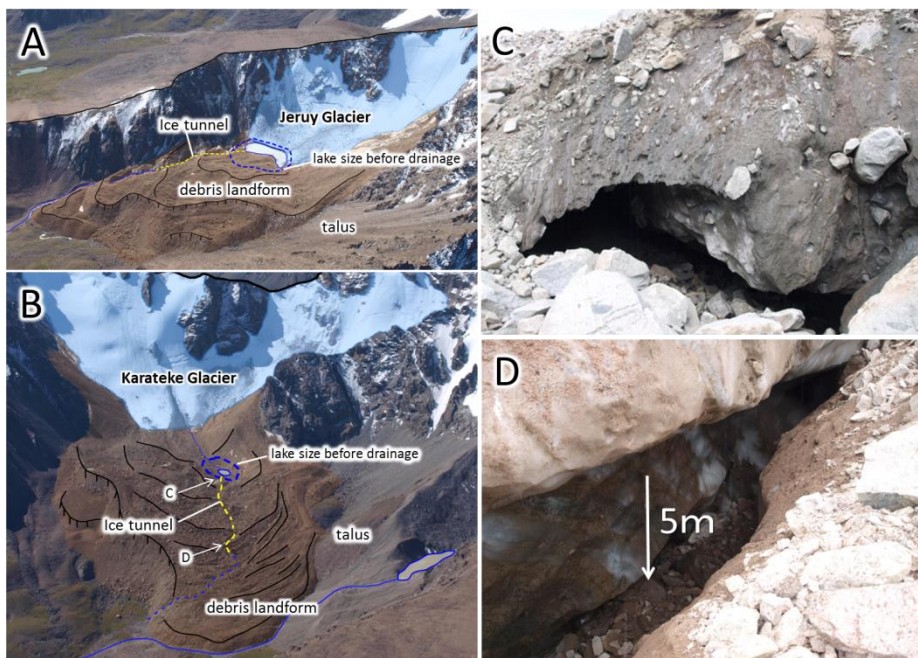


Fig. 5. Surface details of Jeruy and Karateke Glaciers (locations in Fig. 1). In A) and B),
the blue dashed lines show the lake size before drainage. Yellow dashed lines locate
ice-tunnels. C) The ice tunnel in the middle of the debris landform. D) The entry point
of the ice tunnel at the front of Karateke Glacier. The glacial lake with glacier contact
expanded into Jeruy Glacier, and the glacial lake without glacier contact developed in
front of Karateke Glacier.

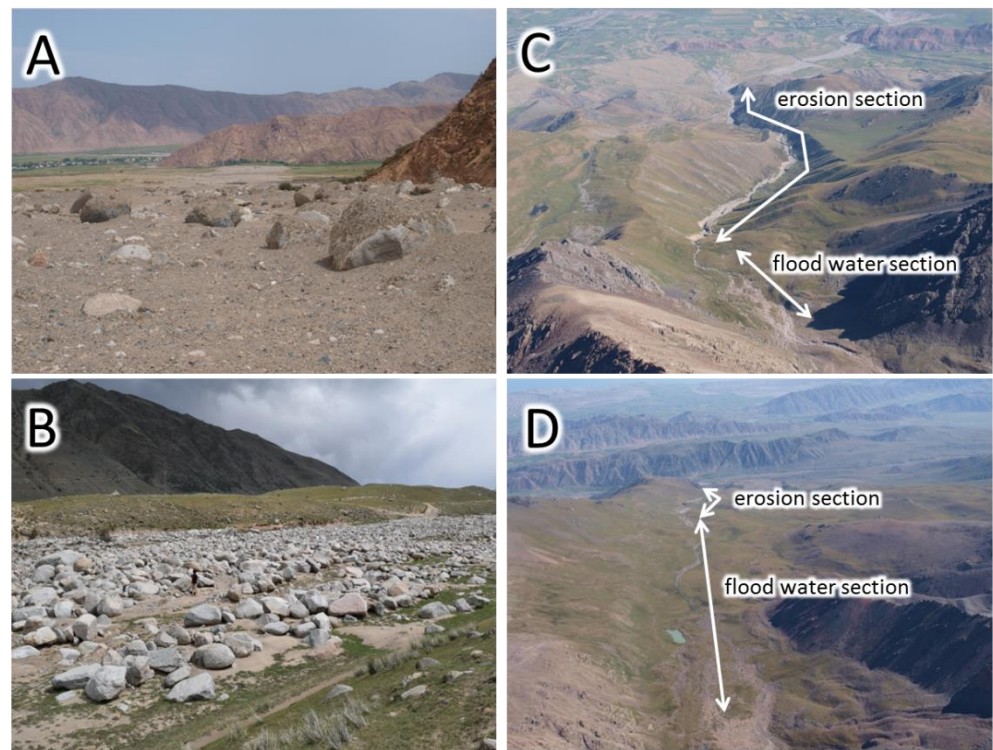

Fig. 6. Flood deposits and valley landforms in the Jeruy (left column) and Karateke
(right column) Valleys.

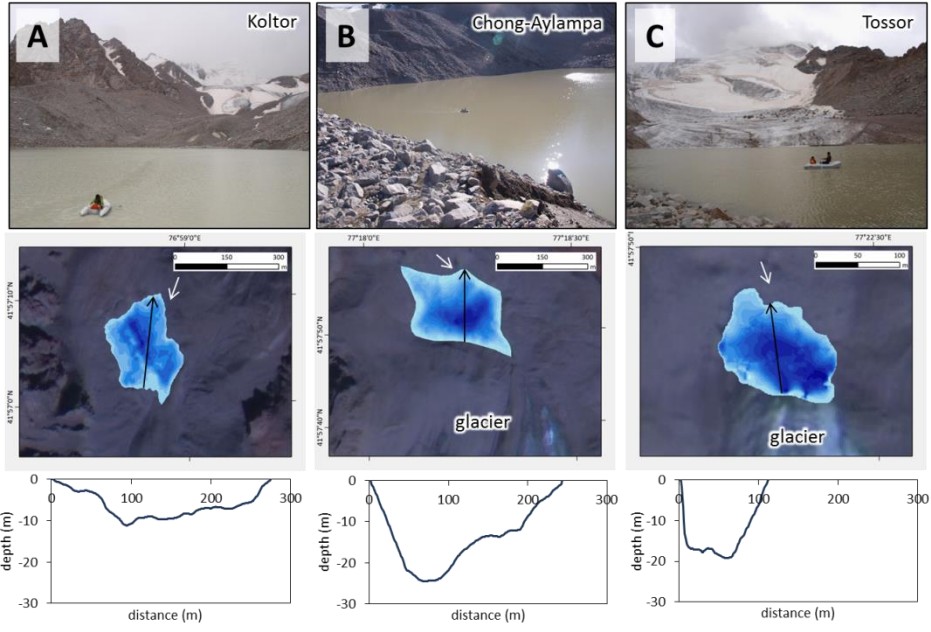

Fig. 7. Photos, lake-basin maps, and depth profiles of three glacial lakes. (A) Koltor, (B)
Chong-Aylampa, (C) Tossor lakes (locations in Fig. 1). Black and white arrows on the
lake-basin maps show each basin profile line and photo direction, respectively.

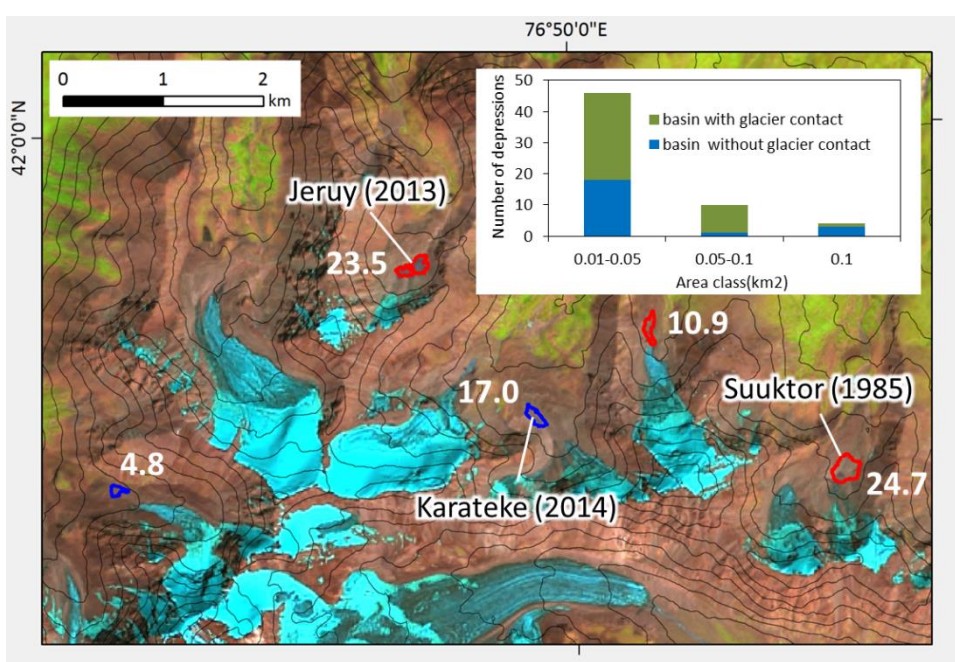


Fig. 8. Depressions at glacier fronts in the study area (locations in Fig. 1). Units are $10^4$
$m^3$. Blue lines outline depressions, and red lines outline depressions having a lake in
2015. The inset shows size, number, and type (with glacier contact or without glacier
contact) of depressions in the study area.

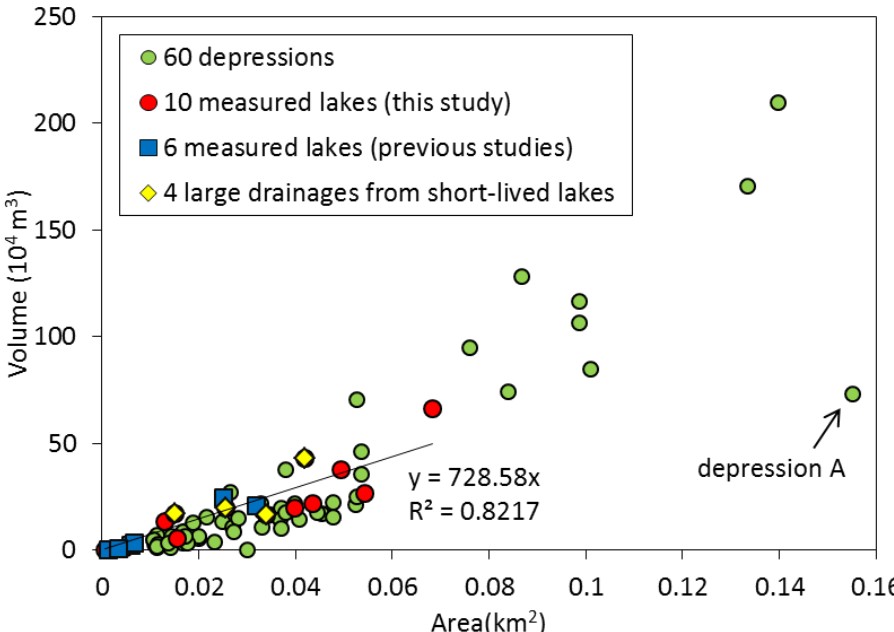


Fig. 9. Volume and area of directly measured lakes and depressions. Fourteen
depressions have an existing glacial lake. Depression A and five depressions have a lake
area of more than 30% of the maximum filling area possible. (Data from 6 previously
studied lakes, this study, and personal communication of I. Severskiy; Janský et al.,

734    2010)

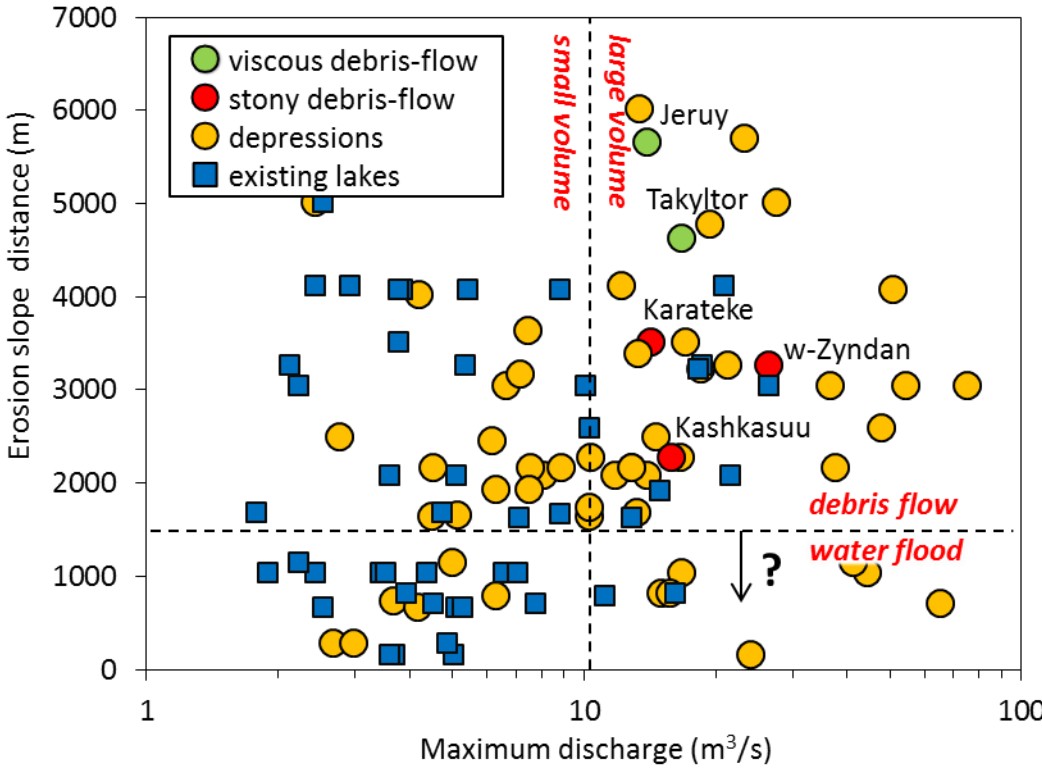



Fig. 10. Debris-flow types in the study area. The horizontal dashed line separates the
debris flows from those with little debris, which we call simply 'water flood'. The
vertical dashed line separates the high-volume flows from the relatively low-volume
flows.
