# Peer review of "Large drainages from short-lived glacial lakes in the Teskey Range,"

_Natural Hazards and Earth System Sciences, 2017_

## Referee Comment (RC1) · Anonymous Referee #1 · 23 Jul 2017

line 95. ... in the interior ... Better "... in the inner Tien Shan ..."

line 145. ... decides. Better ... determines

line 159. Better 2005 and remains the same by 23 May 2006

line 167. ... volume of 163.000 m3

line 187-188. ... we observed 300 m long ice tunnel with ...

line 192-193 Karateke lake also was formed at an empty lake-basin depression, but without glacier contact.

[Figure]

End of section 4.2. Dense debris blows could be quite mobyle too. During the experiments in Kazakhstan in 1972-73 mean density of highly mobyle debris flow reached 2200 kg/m3 and water content was less than 10%

line 241 ... had no such barriers.

line 268. ... short-leaved water bodies (do not use "glacial lakes" twice in one phrase

line 269. ... revealed that water discharged ...

line 273-274 . ... or by deposition of ice and debris ...

line 294 and 534. may be Shatravin?

line 346-348. As these deposits have characteristics of both matrix support and clast support, we treat the flow as viscous.

line 365-366. ... becase banks of the channes in the study area are composed of loose material.

line 386 Many lake-basin depressions are of stony type. It is unclear how they can retain lakes - stony banks and bottom must be permeable.

lines 402-403 - unclear statement.

Figure 4. What is 4/1, 5/1 ...month/day? better to indicae directly.

Figure 6. I do not see dashed region mentioned in the caption.

Figure 8. You indicate "lake-basin with large lake", But only its area is large while volume is small.

---

## Referee Comment (RC2) · A. Emmer (Referee) · 9 Aug 2017

Review to the manuscript 'Large drainages from short-lived glacial lakes in the Teskey Range, Tien Shan Mountains, Central Asia' submitted by Chiyuki Narama et al. to the Natural Hazards and Earth System Sciences. I'm convinved that this manuscript might be of interest for readers of this journal.

General comments:

1) **Structure of the study.** Authors present original field data, analysis of RS data and outline implications for RS-based hazard identification. In order to make the manuscript more readable, I suggest to clearly separate: (i) descriptive part (past events; detection of potential lake basins, ...); (ii) implications for hazard identification and monitoring (geomorphological criterias for hazard identification, recommendations for monitoring, ...). These are mixed up in the current version of the manuscript.

2) **Terminology and language.** Some of the terms used sound bit unusual / clumsy to me, e.g., 'ice-containing debris-landform' – in this case I suggest to use 'ice-cored moraine complex'. Similarly, 'glacier-contact type' might be replaced by 'proglacial' and 'without glacier contact' by 'glacier-detached' (see also Emmer et al., 2015a); 'lake-basin depression' also sounds weird, what about to use 'potential lake basin' or simply 'depression' or 'hollow'; see also my specific comments; this should be unified within whole manuscript. Also, check correct use of present and past tense. Overall, language need some polishing (see also my specific comments).

Specific comments and technical notes:

L18-L19: change wording to 'Four lake drainages from glacial lakes have occurred in the western Teskey Range, Kyrgyzstan, during 2006 and 2014'; check and edit similar cases in whole manuscript

L19: damages

L22-23: four events are mentioned at the beginning of an abstract and three are mentioned here; this is bit confusing and should be edited

L27: late summer

L29: freezing of outflow tunnel

L64-66: fromation and sudden drainage of supraglacial ponds were also documented from the Cordillera Blanca, Peru (see Emmer et al. 2015b)

L67-69: in my understanding, the term 'proglacial' fits there well (e.g., short-lived proglacial lake)

L87-104: please add some info on geological / geomorphological setting of the study area

L114-L117: what is the resolution / accuracy of these measurements

L118-119: please, provide more details on this investigation – what has been done and how ??

L137-138: these images are 7+ years old; considering high dynamic of studied entities, potential outdating should be discussed

L149-150: please, provide more details on this investigation in methodology

L158-162: this is hard to follow; please consider graphical representation of these data

L165: replace 'lies' by 'is located' in entire manuscript

L166: replace 'undiscernible' by 'not recognisable' or something similar

L167: volume 163,000 m$^3$

L169: consider replacing 'non-glacier-contact' by 'glacier-detached' (see Emmer et al., 2015)

L183: adjacent landform

L184: was found

L188-189: how do you know that ?? please provide more info on that

L191: delete 'much'; how much ice ?? please provide more info on that

L202-203: considering the growth of the lakes

L202-204: these growth characteristics better fit in 4.1

L208: ... those that discharge following different mechanism (e.g., dam failure)

L217-220: how was the flow speed in this case ?? please provide more info on that

L222: how much ?? please provide more info on that

L227-228: this is not understandable, please reformulate

L235: why to compare with Himalayas ?? examples from Tien Shan are more reasonable here

L263: does

L266: why is this sub-chapter placed in discussion section ??

L280-282: what process ?? please reformulate / explain

L286: replace 'here' by 'in this study'

L287-288: this might be due to the small events are not documented from Himalayas; please discuss that

L288-289: this is not necessarily true; e.g. in the Cordillera Blanca, most of the lakes which produced GLOF by moraine dam failure still exist

L286: replace 'like' by 'similar to'

L294: this implication is not clear to me, please explain

L302: delete 'much'

L304: Bolch et al. (2011) estimated ??

L311-312: check wording

L312-314: this is not understandable, please reformulate

L319-312: is this shown ?? at the same time, proglacial lake are also turning to glacier-detached phase (see also Emmer et al., 2016); please explain and discuss

L348: replace 'treat' by 'classify'

L362: 'debris-free drainage water', please explain

L363: delete 'the' before observed

L348: replace 'materials' by 'material'

L380-382: discussion or results ??

L383-391: discussion or results ??

L395: what is meant by 'monuments' ?? please explain

L400: term 'landform' is not fitting here well, please reformulate

L406: replace 'within' by 'up to'

L410-11: term 'one package of river basin' please reformulate

L418-419: ... tunnels, as a result of winter freezing ...

L425: (iii) no visible outflow channel

L435-447: these are implications / recommendations, not conclusions, please replace

Fig. 4: please replace the description of x (e.g., April, May, ...)

Fig. 6: please check figure heading (there are no red arrow or dashed region on my figure)

Fig. 8: replace 'prebious' by 'previous'

Fig. 10: replace 'current' by 'existing'

- - -

To sum up, this study needs some improvements before can be published and, based on above mentioned, I suggest medium revisions. I'll be happy to review revised version. Please, do not hesitate to contact me in case of questions (aemmer@seznam.cz).

Kind regards

Adam Emmer

References:

Emmer, A., Merkl, S., & Mergili, M. (2015a). Spatio-temporal patterns of high-mountain lakes and related hazards in western Austria. Geomorphology, 246, 602–616.

Emmer, A., Loarte, E., Klimeš, J., & Vilímek, V. (2015b). Recent evolution and degradation of bent Jatunraju glacier (Cordillera Blanca, Peru). Geomorphology, 228, 345–355.

Emmer, A., Klimeš, J.,  Mergili, M., Vilímek, V., Cochachin, A. (2016): 882 lakes of the Cordillera Blanca: an inventory, classification, evolution and assessment of susceptibility to outburst floods. Catena, 147: 269-279. doi: 10.1016/j.catena.2016.07.032.

---

## Referee Comment (RC3) · Anonymous Referee #3 · 25 Aug 2017

Dear Editor, thank you for allowing me to comment on this manuscript. The Authors proposed the manuscript "Large drainages from short-lived glacial lakes in the Teskey Range, Tien Shan Mountains, Central Asia". The paper is interesting, but Authors have to go through all the text because some sentences are not clearly understandable and need to be reformulated. I recommend it for publication after some medium revision. punteggiatura Main comments: Line 18-23: you first mention four drainages from glacial lakes, but only three are named in the abstract. This happen again further below in the manuscript. That is confusing, please correct it. Line 72: need more references. Line 153: same as line 18-23. Line 155-156: not clear sentence, please rephrase. Line 158: "(left column)" it is not necessary. Line 196: "250 m long". Line

209-220: same as line 18-23. Line 245-246: where are from the data about "four large drainages.... and lake-basin depressions"? Provide references please. Line 247: replace "relational line" with "regression line". Line 249: replace "figure" with "data" (I can not see it from the figure). Line 254-255: merge these two sentences, it look like a repetition. Line 257: "0.01". I suggest to use the same decimal precision (0.015 in other parts of the manuscript). Line 262: not clear sentence, please reformulate (and replace "dose" with "does"). Line 267: "As shown in Fig. 3, consecutive...". Line 310: "2.5 m". Line 311-312: not clear sentence, check wording. Line 315: replace "conditions" with "characteristics". Line 336: please explain (iv). Line 343: "maximum Discharge (Qmax)". Line 356-357: replace "the characterization" with "this classification". Figure 6: there are no dashed region or red arrows in the figure (maybe white).

---

## Author Comment (AC2) · 13 Nov 2017

Dear Referee

We would to thank all referees for their valuable comments and have implemented all of them as attached supplement file. We restructured and changed in discussion part.

You will find attached the revised text/figures and our response for reviewer's comments.

Sincerely, Chiyuki Narama

Please also note the supplement to this comment:
https://www.nat-hazards-earth-syst-sci-discuss.net/nhess-2017-228/nhess-2017-228-AC2-supplement.zip
* * *

---

## Author Comment (AC3) · 13 Nov 2017

Dear Referee

We would to thank all referees for their valuable comments and have implemented all of them as attached supplement file. We restructured and changed in discussion part.

You will find attached the revised text/figures and our response for reviewer's comments.

Sincerely, Chiyuki Narama

[Figure]

Please also note the supplement to this comment:
https://www.nat-hazards-earth-syst-sci-discuss.net/nhess-2017-228/nhess-2017-228-AC3-supplement.zip

---

## Author Response (AR1)

Dear Editor Dr. Filippo Catani,

We sincerely thank you for the efforts you have made to improve our submission to NHESS. We have responded to your comments. The sentences in blue font are your comments, those in black are our responses.

After reading and evaluating the reviews given by the referees, and after my personal check on your last version's manuscript, I guess we are almost finished and that your paper may be considered acceptable provided that you are able to answer a few remaining doubts:

- line 163 - Please provide some basis for the choice of using SRTM (year of acquisition 2000) instead of ALOS DSM (year of acquisition 2007 and 2010). Is the area under investigation outside the ALOS DSM dataset coverage?

We used SRTM DEM in this paper, because we have a plan to use SRTM DEM to calculate erosion distance for spread area of Issyk-Kul Basin. In this basin, the ALOS DSM has some gaps of coverage. This is now mentioned on lines 167-168.

- line 260 - the sentence "... much debris led to slow velocities..." is not clear. Please rephrase

We changed to

"Although the Jeruy drainage has a long eroded section which can indicate a fast flow, the flood here did not include many large boulders. The flow had relatively slow velocity on the alluvial fan."

- lines 300-306 - Could you add some considerations on cases that do not fall on or near the regression line in Fig. 9? You use eq. to compute volumes but what about cases where the fit is not good?

We added the following explanation:

"The larger-area lakes have volumes above the regression-line fit because those depressions tend to have steeper sides. Although we could use a second-order fit to better fit these points, our subsequent analyses below all use lakes with areas below 0.05 km2. Thus, we use the simpler first-order relation."

"

I will personally review the corrections that you will make, without recurring to a new review round.

I am confident that this will greatly expedite the publication process.

Thank you for choosing NHESS Journal.